# Homogenization of Extrusion Billets of a Novel Al-Mg-Si-Cu Alloy with Increased Copper Content

**DOI:** 10.3390/ma16052091

**Published:** 2023-03-03

**Authors:** Antoni Woźnicki, Beata Leszczyńska-Madej, Grzegorz Włoch, Jacek Madura, Marek Bogusz, Dariusz Leśniak

**Affiliations:** 1Aptiv Services Poland S.A., 30-399 Kraków, Poland; 2Faculty of Non-Ferrous Metals, AGH University of Science and Technology, 30-059 Kraków, Poland

**Keywords:** Al-Mg-Si-Cu alloys, extrusion billets, homogenization soaking, cooling from homogenization

## Abstract

Within the present work the homogenization of DC-cast (direct chill-cast) extrusion billets of Al-Mg-Si-Cu alloy was investigated. The alloy is characterized by higher Cu content than currently applied in 6xxx series. The aim of the work was analysis of billets homogenization conditions enabling maximum dissolution of soluble phases during heating and soaking as well as their re-precipitation during cooling in form of particles capable for rapid dissolution during subsequent processes. The material was subjected to laboratory homogenization and the microstructural effects were assessed on the basis of DSC (differential scanning calorimetry) tests, SEM/EDS (scanning electron microscopy/energy-dispersive spectroscopy) investigations and XRD (X-ray diffraction) analyses. The proposed homogenization scheme with three soaking stages enabled full dissolution of Q-Al_5_Cu_2_Mg_8_Si_6_ and θ-Al_2_Cu phases. The β-Mg_2_Si phase was not dissolved completely during soaking, but its amount was significantly reduced. Fast cooling from homogenization was needed to refine β-Mg_2_Si phase particles, but despite this in the microstructure coarse Q-Al_5_Cu_2_Mg_8_Si_6_ phase particles were found. Thus, rapid billets heating may lead to incipient melting at the temperature of about 545 °C and the careful selection of billets preheating and extrusion conditions was found necessary.

## 1. Introduction

The automotive branch, which is one of the main aluminum alloys consumers, undergoes currently significant transformation resulting from walk towards electromobility. Between year 2018 and 2021, the share of electrically chargeable vehicles (i.e., battery electric vehicles and plug-in hybrids) among new cars registered in the European Union increased from 1.9 to 18% [1]. Considering announced legal restrictions for cars with internal combustion engines one can expect further growth of electric cars share in markets worldwide. This change is an origin of challenges for suppliers of aluminum alloys components, which are used in electric cars. As an example, one may show battery enclosures, in which aluminum alloys extrusions are often used [2,3]. The enclosure must ensure the protection of heavy battery from crashes and strikes as well as enable a thermal management. To provide this, the high-strength extruded profiles with complex shapes, often hollow, are needed. It is worth noticing that high-strength, light components are generally desired in electric cars. Battery increases noticeably vehicle mass, hence there is a need of reducing other components mass, which is very important in terms of range.

The requirement of complex profiles shape causes that among precipitation hardenable aluminum alloys, the 6xxx alloys, in which high strength properties can be combined with good extrudability, are considered as materials of choice. The high-strength 6xxx alloys often have Cu as an essential addition, with its concentrations up to 1.2 wt% [4]. Many of mentioned alloys contain the quaternary Q-phase. Its exact composition is unknown and is reported as Al_5_Cu_2_Mg_8_Si_6_, Al_4_CuMg_5_Si_4_, Al_4_Cu_2_Mg_8_Si_7_, and Al_3_Cu_2_Mg_9_Si_7_. In equilibrium state the Q phase can coexist with (Al) solid solution and two of other three phases: θ-Al_2_Cu, β-Mg_2_Si, and (Si) [5]. After ageing of Al-Mg-Si-Cu alloys the significant strengthening effects may arise from precursor phases to Q’ [5,6]. Despite the fact that Al-Mg-Si-Cu alloys are used since 1950s, since this time grades characterized by tensile strength in T6 temper close to 500 MPa have been elaborated, e.g., 6069 [7], and there are other interesting subjects for investigation in the field of Al and its alloys, e.g., [8,9,10], the considered alloys are still object of researchers’ interest. The works are related to modification of alloys chemical composition as well as application of tailored processing parameters [11,12,13,14].

In the case of the Al-Mg-Si-Cu alloys, a solidification comprises several reactions and the microstructure in as-cast state is complex [15,16,17]. Between (Al) dendrites arms numerous phases can be observed: soluble β-Mg_2_Si, θ-Al_2_Cu, Q and (Si)—in some cases all of them are reported—as well as Fe-bearing phases e.g., α-Al(FeMn)Si, β-Al(FeMn)Si. The unequilibrium solidus temperature may be even as low as about 485 °C [16].

The homogenization of DC-cast billets is an important part of the extruded products manufacturing cycle, significantly influencing billets extrudability as well as properties of obtained profiles. The primary microstructural process occurring during homogenization heating and soaking is dissolution of soluble phases particles formed during solidification and elimination of microsegregation. At the end of the soaking stage, a maximum attainable enrichment of (Al) solid solution in Mg, Si, and Cu, taking part in subsequent precipitation hardening, shall be obtained. A high soaking temperature facilitates the particles dissolution. However, the low-melting microstructure components (eutectics) present in the as-cast billets need to be removed during the heating stage. In some cases, an intermediate soaking at low temperature (470 °C) is applied, after which a final soaking at high temperature (560–570 °C), may be safely accomplished [16,18].

In 6xxx alloys containing transition metals, usually Mn and/or Cr, during homogenization heating and soaking precipitation of dispersoids takes place. These particles have strong effect on recovery, recrystallization and grain growth processes during hot working and annealing treatments. However, to achieve expected results, the dispersoids need to be fine, densely and uniformly distributed. Thus, the homogenization parameters of high strength 6xxx alloys must be selected with respect to these particles. A slow heating to homogenization or application of intermediate soaking step in course of heating are reported to positively influence dispersoids distribution [19,20,21,22]. If final homogenization soaking at high temperature is applied, a stronger tendency of the dispersoids to coarsen is observed, when compared to low temperature soaking, e.g., 560 vs. 500 °C in [11]. On the other hand, low soaking temperature, favorable for dispersoids, may lead to insufficient dissolution of phases, components of which take part in subsequent precipitation hardening [23].

Another microstructural processes occurring during 6xxx alloys homogenization are related to Fe-bearing phases formed during solidification: transformation of undesirable β-Al(FeMn)Si phase into α-Al(FeMn)Si, changes in particles composition and/or morphology. These processes influence the billets extrudability and quality of extruded products surface [24,25,26].

After completed soaking, the extrusion billets are cooled to the ambient temperature. Within this stage, the phases showing limited solubility in Al matrix re-precipitate. It is of great importance to ensure that the phases will be present in billets structure in form of particles, which are capable for fast dissolution during subsequent billets preheating and extrusion. If the cooling is too slow, the obtained particles are too large to fully dissolve. It results in reduced concentration of alloying additions in (Al) solid solution and in consequence in lowered strength properties of profiles after subsequent quenching on the press output and ageing. Moreover, presence of undissolved particles in the billets microstructure causes an incipient melting of the alloy at a relatively low temperature, as the result of the unequilibrium eutectic reaction between the particles and surrounding matrix. In order to avoid melting, leading to profiles surface defects appearance, the exit temperature of the profile must be kept below the eutectic temperature and the permissible extrusion speed is reduced [27,28]. On the other hand, too high cooling rates from homogenization temperature are also unfavorable, because of rising supersaturation of the solid solution with alloying additions. It increases flow stress of the material and breakthrough-pressure [29]. The process of cooling billets of Al-Mg-Si alloys is often described in the literature, e.g., [30,31,32], but there is a lack of similar analyses of 6xxx alloys containing Cu.

The literature data presented above may be summarized as follows: The expected microstructure of homogenized billets is characterized by the presence of soluble phases in the form of fine particles, capable for rapid dissolution during billets preheating and extrusion. In order to achieve this, the homogenization soaking should ensure the maximum possible dissolution of mentioned phases. Due to low melting temperature of the as-cast billets, a multi-step homogenization may be necessary. The conditions of cooling of Al-Mg-Si-Cu billets leading to the refinement of particles precipitated during cooling need to be investigated.

This paper presents the results of investigations of new 6xxx alloy, with higher Cu content than currently applied, intended for application in automotive industry. The work focuses on the analysis of billets homogenization conditions leading to achieving the microstructure as described above.

## 2. Materials and Methods

The material for investigation came from billet with a diameter of 100 mm, cast in semi-industrial conditions with direct-chill method. The chemical composition of the alloy is presented in Table 1. From the obtained billet, the cuboid specimens with dimensions of 10 × 20 × 15 mm were sectioned. They were intended for examination in the as-cast state as well as for laboratory homogenization experiments.

At the first stage of work, the material in the as-cast state was subjected to DSC (differential scanning calorimetry) tests, XRD (X-ray diffraction) analyses, as well as microstructure observations. The DSC tests were performed using a Mettler Toledo 821^e^ heat flux type calorimeter (Greifensee, Switzerland). The disc-shaped samples were inserted in ceramic pans into the cell with the temperature of 390 °C and heated at 20 °C/min to the temperature of 700 °C in Ar atmosphere. The solidus temperature as well as heat of the incipient melting reactions were determined. The incipient melting reactions heat is given in J/g—the heat value determined from DSC curve was divided by whole specimen mass.

The phase composition of the powdered specimens was analyzed using the Bruker D8 Advance (Karlsruhe, Germany) X-ray diffractometer with Co Kα = 1.79 Å. The analyses were performed in a 2θ angle range of 15° to 100°. 

The specimens for microstructure examination were mounted in conductive resin, subsequently mechanically ground and polished using abrasive papers, diamond suspensions and colloidal silica suspension. The billets microstructure was examined using SEM/EDS (scanning electron microscopy/energy-dispersive spectroscopy). The SEM/EDS analyses were performed on non-etched specimens using Zeiss Sigma 300 VP scanning electron microscope (Jena, Germany) equipped with an Oxford Instruments EDS system (Abingdon, England) as well as a Hitachi SU-70 scanning electron microscope (Tokyo, Japan) equipped with a Thermo Scientific EDS system (Waltham, MA, USA). The EDS analyses were applied to determine the chemical composition of the observed eutectic areas or particles and to measure the main alloying elements concentration in the dendrites interiors.

At the second stage of work, soaking parameters were investigated. The homogenization scheme with three soaking stages at 475, 530, and 545 °C was applied (Figure 1a). The temperature of first soaking was selected on the basis of the literature data [16,18,20] and the results of DSC tests of alloy in the as-cast state showing the solidus (incipient melting) temperature. The temperatures of second and third soaking steps were set at about 15 °C below the solidus temperature obtained after preceding soaking. The specimens were heated from room temperature to 475 °C for 10 h. A similar heating rate, about 40 °C/h, was applied during heating between further soaking stages. After completed soaking, specimens were quenched in water. The heat treatment experiments were accomplished using a Nabertherm forced convection chamber furnace.

At the third stage, the influence of cooling rate from the homogenization temperature on the billet microstructure was studied. Specimens were subjected to homogenization with soaking conditions selected on the basis of second stage results, i.e., 475 °C/2 h + 530 °C/2 h + 545 °C/8 h. Then they were cooled to room temperature in three ways (Figure 1b). The average cooling rates in the temperature range from 545 to 200 °C, estimated on the basis of specimens temperature measurements during cooling cycles, were about 500, 300, and 50 °C/h.

Specimens after all homogenization experiments were subjected to DSC tests. In the second stage, the dissolution of low-melting microstructure components was analyzed. In the third stage, the ability of precipitated particles to dissolution during rapid heating was evaluated. On the basis of the obtained DSC results, specimens for XRD analyses and microstructure observations were selected. The examination of homogenized alloy was performed in the manner described above.

The backscattered electron images of specimens cooled after homogenization with different rates were additionally subjected to image analysis procedure. For each specimen, six photographs at magnification of 1000× were acquired in randomly selected areas. Microstructure images were binarized in order to extract β-Mg_2_Si particles. Their number and the surface area of particles cross-sections were measured using image analysis software Struktura 1.0, developed at AGH-UST. Based on the obtained surface area, the equivalent diameter of particles cross-sections was calculated and used for particles classification. In the text below, descriptions of particles dimensions or area refer to values determined from observed particles cross-sections. 

Similar procedure was not applied to other soluble phases, Q-Al_5_Cu_2_Mg_8_Si_6_ and θ-Al_2_Cu, due to their poor contrast with particles Al(FeMn)Si.

## 3. Results

### 3.1. DSC

On the DSC curves in as-cast state, three peaks resulting from incipient melting reactions are observed (Figure 2a). First, denoted as A, has an onset at the temperature of 513 °C, this is unequilibrium solidus temperature of investigated billet. Second peak (B) has an onset at about 539 °C, third peak (C) at about 561 °C. The melting heat values are 0.47, 1.48, and 2.12 J/g respectively (Table 2). 

After heating to the temperature of 475 °C, soaking for 2 h and water quenching, peak A is not observed and the solidus temperature rises to about 545 °C. Heat of peak B decreases to about 0.9 J/g, whereas heat of peak C rises to 3.8 J/g. The subsequent heating to the temperature of 530 °C and soaking for 2 h, causes the peak B to vanish and the solidus temperature to rise to about 561 °C. Heat of the peak C decreases to about 2.4 J/g. Owing to the solidus temperature increase another soaking step, at 545 °C, could be accomplished. After 4 h of soaking, the peak C is merged with bulk melting peak, and after 8 h followed by water quenching, it vanishes and solidus temperature rises to about 573 °C (Figure 2a, Table 2).

The DSC curves after three-stage homogenization with final soaking at 545 °C for 8 h and different cooling manners are shown in the Figure 2b. After cooling at 50 and 500 °C/h, on the DSC curves two peaks are observed. First peak has an onset at a temperature of 543–547 °C and its heat is very small: 0.02 and 0.1 J/g respectively for 50 and 500 °C/h (Table 2). After cooling at 300 °C/h this peak is not observed. Second peak has an onset at a temperature 559–563 °C and is visible after all cooling manners. Heat of this peak decreases with cooling rate, values of 5.95, 0.78, and 0.14 J/g are obtained for 50, 300, and 500 °C/h respectively (Table 2). The mentioned peaks correspond to peaks B and C described for as-cast state and during soaking conditions analysis.

### 3.2. SEM/EDS

The investigated alloy in as-cast state was characterized by dendritic microstructure with numerous particles and eutectic areas located at dendrites/grains boundaries (Figure 3). Microanalyses of the observed particles indicate the presence of phases: β-Mg_2_Si, phase containing Al, Mg, Si, and Cu and phase containing primarily Al, Mn, and Si, with Fe, Cr, and Cu also detected. The second mentioned phase was identified as Q-Al_5_Cu_2_Mg_8_Si_6_, it was done on the basis of spectra quantification results showing that Cu:Si ratio, expressed in at. %, is close to 1:3 [16]. The Fe, Mn, and Si-bearing particles are probably α-Al(FeMn)Si phase, their composition is very close to the particles of this phase described in [12]. The eutectic areas are often multicomponent and the EDS microanalyses results show that besides (Al) they are composed of phases Q-Al_5_Cu_2_Mg_8_Si_6_, θ-Al_2_Cu, and (Si) (Figure 3). The dendrites interiors are noticeably impoverished in main alloying additions, Mg, Si, and Cu (Figure 6).

After heating to 475 °C for 10 h and soaking for 2 h, fraction of particles in the microstructure noticeably decreases (Figure 4a). The β-Mg_2_Si, Q-Al_5_Cu_2_Mg_8_Si_6_, and Al(FeMn)Si particles are observed, phases θ-Al_2_Cu and (Si) are not found. At higher magnification, in grains interiors dispersoids are visible. Second soaking step, at 530 °C for 2 h, causes dissolution of phase Q-Al_5_Cu_2_Mg_8_Si_6_, particles β-Mg_2_Si and Al(FeMn)Si are present (Figure 4b). After final soaking, the same particles are observed as after the second one, but amount of β-Mg_2_Si is smaller (Figure 4c). The observations at high magnification after second and third soaking reveal a presence of small, elongated particles, bright in the backscattered electron images. Microanalyses show that the particles are rich in Cu and Si, example is shown in the Figure 5. Fraction of this particles in the microstructure can be assessed as very small.

One should notice that after complete soaking, the dispersoids present in the microstructure are uniformly distributed and the tendency of them to coarsen is not found—dimensions of dispersoids are far below 1 μm. The particles of Al(FeMn)Si phase underwent fragmentation and their edges became more rounded (Figure 4c and Figure 5). 

The concentration of Mg, Si, and Cu in grains interiors rises in line with particles dissolution (Figure 6). After first soaking stage at 475 °C the concentration of Si and Cu increases noticeably, when compared to as-cast state, which is related to dissolution of θ-Al_2_Cu and (Si) observed in eutectic areas, probably to some extent also with dissolution of Q-Al_5_Cu_2_Mg_8_Si_6_ phase. The second soaking, at 530 °C, causes an increase in all main alloying additions concentration, resulting from of Q-Al_5_Cu_2_Mg_8_Si_6_ phase dissolution. After final soaking at 545 °C, the Cu content in grains interiors remains unchanged and concentrations of Mg and Si rise due to reduction of β-Mg_2_Si phase fraction.

Cooling from homogenization temperature at 50 °C/h results in precipitation of coarse particles (Figure 7 and Figure 8). The particles of β-Mg_2_Si phase are rather equiaxed, their dimensions are often above 3 μm. Particles of Q-Al_5_Cu_2_Mg_8_Si_6_ phase are most often elongated. Their length is even above 10 μm and thickness of largest is between 1 and 2 μm. Particles of θ-Al_2_Cu phase are also elongated, but noticeably smaller than Q-phase, they have length up to several μm and thickness below 1 μm. With the increase in cooling rate, as expected, the number of large β-Mg_2_Si particles clearly decreases and mainly finer ones, with dimensions below 2 μm are observed in the microstructure, although few large particles are also noted. The increase in cooling rate results in lowering of total area of precipitated particles, the (Al) matrix becomes to some extent supersaturated. The θ-Al_2_Cu particles also become finer. In the case of Q-Al_5_Cu_2_Mg_8_Si_6_ phase, results are unexpected. On the one hand the rising cooling rate causes precipitation of fine particles in grains interiors, but on the other hand even after cooling at 500 °C/h large and thick particles are still easily observed. After all examined cooling manners, dimensions of the unfavorable, largest Q-Al_5_Cu_2_Mg_8_Si_6_ particles found in the microstructure are very similar.

### 3.3. XRD

XRD spectra for alloy in as-cast state as well as after examined soaking steps are shown in Figure 9. In the spectrum obtained in as-cast state peaks from phases β-Mg_2_Si, Q-Al_5_Cu_2_Mg_8_Si_6_, Si and θ-Al_2_Cu are identified; therefore the above presented identification of these phases on the basis of microanalyses results is confirmed. One should note that strong peaks from expected phases often overlap with peaks from Al or from other phases and the analysis of obtained spectra is rather difficult. After first homogenization stage, at 475 °C for 2 h, peak from β-Mg_2_Si at 47° slightly grows; during further soaking steps it decreases.

## 4. Discussion

The examined alloy after DC-casting shows dendritic microstructure with depleted (Al) solid solution and large amount of particles and eutectic areas at the dendrites boundaries. Three peaks visible on the DSC curve indicate incipient melting reactions. Based on the presented microstructure observations and the literature data following attempt of reactions identification may be proposed: The peak A, with onset at about 513 °C, can be related to melting of eutectic regions in reaction:(Al) + (Si) + Al_2_Cu + Al_5_Cu_2_Mg_8_Si_6_ ⇒ L(1)

In the quaternary Al-Mg-Si-Cu system, this reaction is described to take place at the temperature of 507 °C [33], slightly lower than peak onset temperature. One should note that in [15] the same reaction in commercial alloy 6111 is described to take place at significantly lower temperature, 487 °C.

The peak B in as-cast state has an onset at about 540 °C, and at 545 °C after first soaking at 475 °C. It may be expected that it is related to melting reaction or reactions with the participation of Q-Al_5_Cu_2_Mg_8_Si_6_ phase:(Al) + Al_5_Cu_2_Mg_8_Si_6_ + (Si) ⇒ L(2)
(Al) + Al_5_Cu_2_Mg_8_Si ⇒ L(3)

It is hard to state unambiguously which of the above mentioned reactions causes peak B occurrence in the examined alloy. However, based on (Si) presence in the eutectic areas as well as slightly lower peak B onset in as-cast state, one may expect that in this state peak is related to reaction (2). After first soaking, (Si) was not found in the microstructure, also probably reaction (3) takes place in this case. According to [17], in quaternary alloy Al—1.0 wt.% Mg—1.1 wt.% Si—0.65 wt.% Cu, these reactions took place at the temperature ranges 519–537 °C and 537–539 °C respectively. Moreover, temperature range for these reactions from quaternary phase diagram is noticeably lower: 507–529 °C [33].

The peak C in as-cast state has an onset at 561 °C. During homogenization it rises, finally is merged with bulk melting peak, and the solidus temperature is 573 °C (Figure 2a, Table 2). One may suppose that this peak is related to melting reaction with the participation of Mg_2_Si phase particles and its onset rises due to changes in (Al) solid solution composition. This is based on the microstructure observations showing the presence of Mg_2_Si and Al(FeMn)Si phases, and the literature data indicating that melting of Fe-bearing phases is observed at a higher temperature range [15,33]. 

The maximum possible dissolution during homogenization of phases contributing to precipitation hardening is essential for obtaining high strength properties of extruded products. This requires high temperature soaking, but due to the presence of three incipient melting peaks on the DSC curve, soaking at 545 °C could be applied after two intermediate soaking stages. The applied soaking enables full dissolution of Q-Al_5_Cu_2_Mg_8_Si_6_, and θ-Al_2_Cu phases resulting in enrichment of solid solution in Cu (Figure 6). Some Cu is detected also in Al(FeMn)Si particles as well as in particles rich in Cu and Si. It is hard to describe the nature of the latter particles. The EDS spectra clearly show peaks from Cu and Si. The ternary phases are not present in aluminum corners of Al-Cu-Si and Al-Mg-Si-Cu phase diagrams. On the other hand, the solubility of Si in θ-Al_2_Cu phase is negligible [33]. It is also hard to state if these particles were formed during billet casting or during homogenization. Some microanalyses of eutectic areas constituents in as-cast state showed enrichments in Cu and Si, but it may result from the fact that the volume excited by electron beam is larger than the analyzed eutectic constituent and the X-rays are collected also from neighborhood. These particles should be the subject of further investigations.

Despite the soaking at 545 °C for 8 h, the β-Mg_2_Si phase was not dissolved completely, but its amount was significantly reduced. These results indicate that full dissolution of this phase is very difficult or impossible for the examined composition. Rising of homogenization temperature could be possible in laboratory conditions, but application of higher temperature in industrial practice could create the risk of incipient melting (for example due to temperature variation in the furnace). Considering that for the examined composition some part of Mg and Si will not take part in precipitation hardening, a slight reduction in these elements content during further investigations is justified.

As it was already mentioned, the billets cooling from homogenization temperature should cause refinement of precipitated particles making them capable for fast dissolution. As expected, the rising cooling rate contributed to refinement of Mg_2_Si particles. After cooling from homogenization at 500 °C/h the precipitated particles dissolved nearly completely within short time (about 9 min) during the DSC test. Although on the DSC curve, peak C was noted and solidus temperature was slightly lower than after water quenching, a small heat of the peak C shows that fraction of remaining particles, which could be dissolved, is very low. Even if in industrial conditions such rapid billet heating was applied, the decrease in solidus temperature and depletion of Mg and Si in (Al) would be rather negligible. The particles of θ-Al_2_Cu phase precipitated during cooling are small and do not cause incipient melting. Unexpectedly, even after the fastest examined cooling, in the alloy microstructure coarse particles of Q-Al_5_Cu_2_Mg_8_Si_6_ phase are observed besides fine ones. The significant variation in Q-phase particles size is hard to explain. The presence of coarse particles lowers the solidus temperature of homogenized alloy to about 545 °C. This is unfavorable because the range of permissible exit temperature during extrusion needs to be noticeably reduced in order to avoid melting. However, the heat of peak B is very small (it is hardly visible on the DSC curves), which indicates that amount of undissolved Q-Al_5_Cu_2_Mg_8_Si_6_ phase particles at the temperature of about 545 °C (incipient melting) is very small. Thus, one may expect that in industrial process, where billets heating rate will probably be lower and the plastic deformation during extrusion will facilitate diffusion, these particles will be fully dissolved. Nevertheless, this supposition needs to be verified during extrusion trials and conditions of billets preheating and extrusion need to be selected carefully. The results allow to recommend for the examined alloy fast cooling from homogenization temperature, at least at 500 °C/h.

Within future work, the evaluation of proposed homogenization conditions is planned. The homogenized billets are to be extruded in semi-industrial conditions with solution heat treatment on the press. The extruded products, after ageing, will be subjected to detailed examination of microstructure aimed primarily at verification if undissolved particles of phases, components of which are essential in subsequent precipitation hardening, are present. The solidus temperature will be determined and compared with the value obtained after homogenization with water quenching (within stage 2 of the present work). The mechanical properties tests of extruded products will also be performed.

## 5. Conclusions

On the basis of the obtained results, the following conclusions can be drawn: The examined alloy in as-cast state was characterized by dendritic microstructure with large amount of particles and eutectic areas at the dendrites boundaries. Besides (Al) solid solution, in the microstructure phases β-Mg_2_Si, Q-Al_5_Cu_2_Mg_8_Si_6_, θ-Al_2_Cu, (Si) and Al(FeMn)Si were found. On the DSC curves three peaks resulting from incipient melting reactions were observed, solidus temperature of the alloy in as-cast state was 513 °C.In order to avoid incipient melting, the homogenization scheme with three soaking stages: at 475, 530, and 545 °C was applied. During soaking, full dissolution of Q-Al_5_Cu_2_Mg_8_Si_6_ and θ-Al_2_Cu phases took place. The β-Mg_2_Si phase did not dissolve completely during soaking, but its amount was significantly reduced.The alloy needs to be fast cooled from homogenization temperature, at least at 500 °C/h. The β-Mg_2_Si and θ-Al_2_Cu particles precipitated during cooling were sufficiently refined. Despite such fast cooling, in the microstructure coarse Q-Al_5_Cu_2_Mg_8_Si_6_ phase particles were found and rapid billets heating may lead to incipient melting at the temperature of about 545 °C. Thus, the careful selection of billets preheating and extrusion conditions was found to be necessary.

## Figures and Tables

**Figure 1 materials-16-02091-f001:**
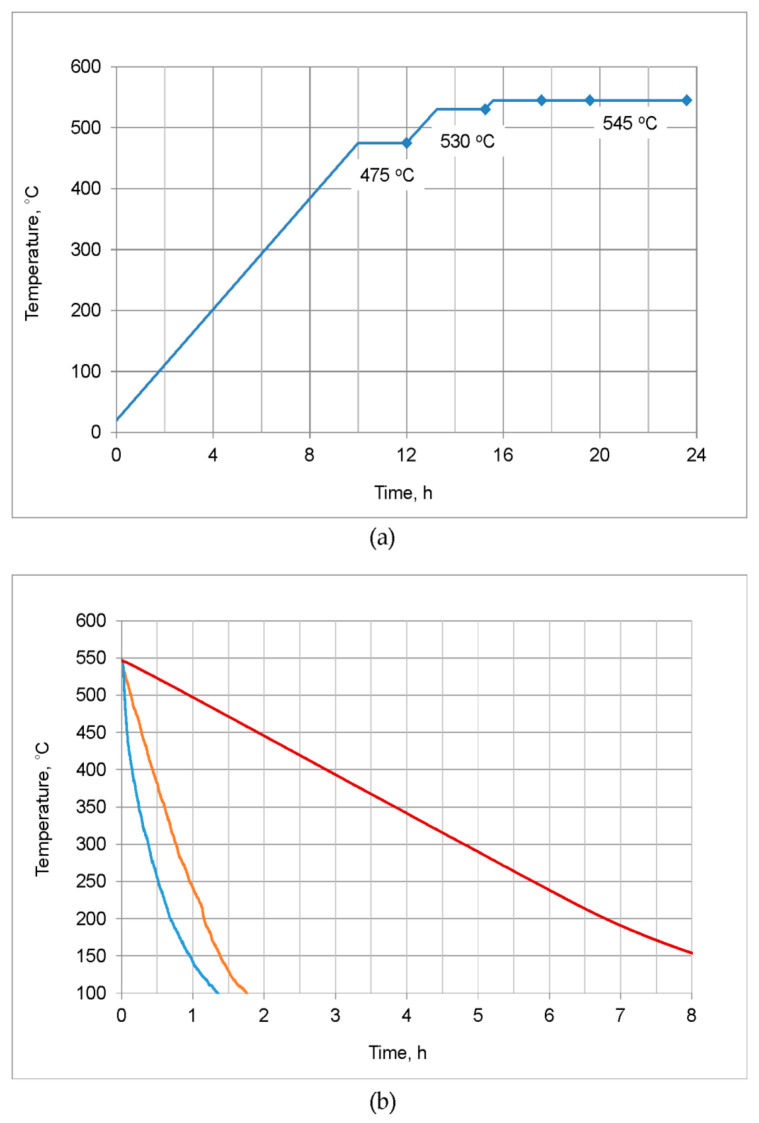
Laboratory homogenization experiments: (**a**) Heating and soaking conditions with marked sampling points (2nd stage of work); (**b**) cooling curves obtained after homogenization with soaking 475 °C/2 h + 530 °C/2 h + 545 °C/8 h (3rd stage of work). The average cooling rates in the temperature range from 545 to 200 °C, were about 500 (blue line), 300 (orange line), and 50 °C/h (red line).

**Figure 2 materials-16-02091-f002:**
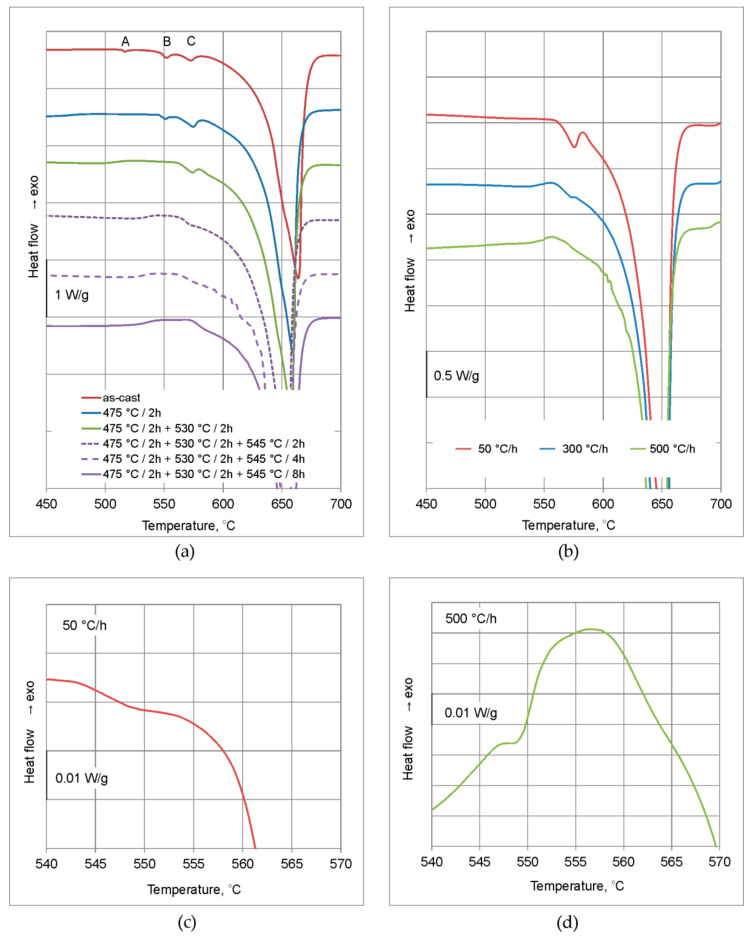
DSC curves: (**a**) As-cast state and after homogenization with differentiated soaking and water quenching; (**b**) after homogenization with soaking 475 °C/2 h + 530 °C/2 h + 545 °C/8 h and differentiated cooling rate; (**c**,**d**) magnified parts of curves from panel **b** showing very small peaks at about 545 °C.

**Figure 3 materials-16-02091-f003:**
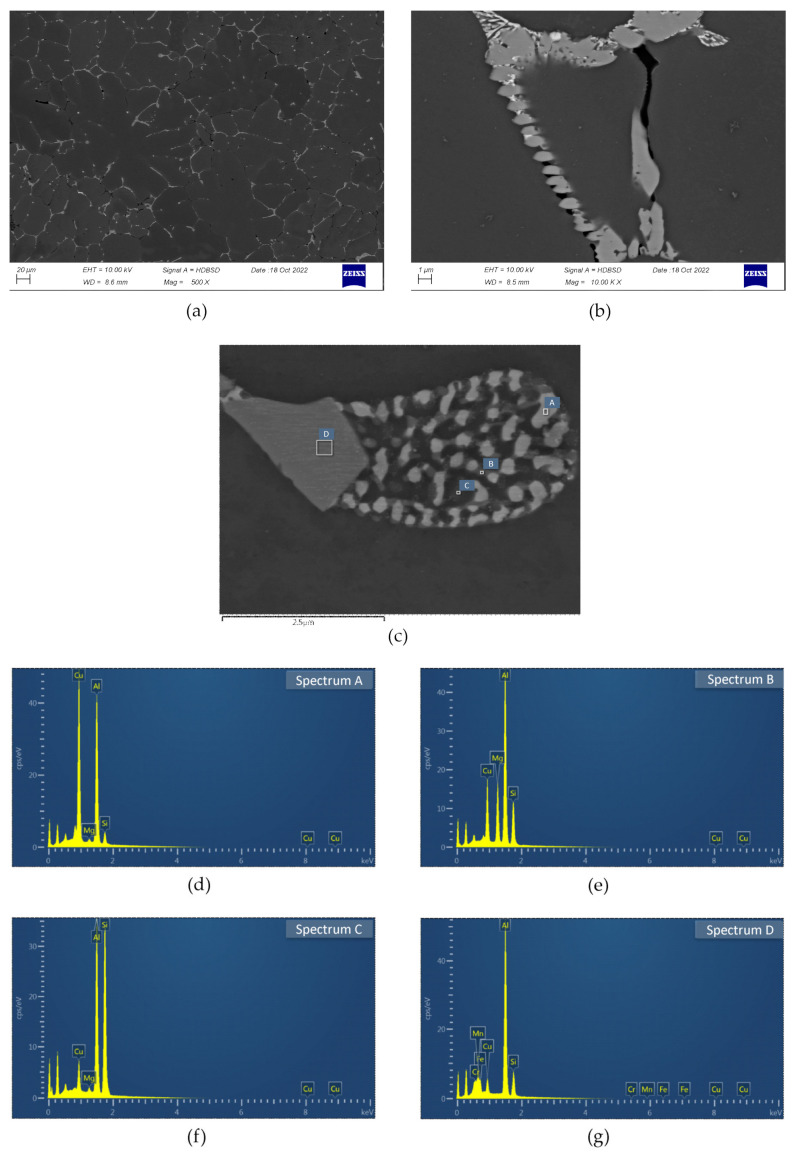
Microstructure of investigated alloy in as-cast state: (**a**) General view of the dendritic microstructure; (**b**,**c**) particles and eutectic areas; (**d**–**g**) spectra from panel c.

**Figure 4 materials-16-02091-f004:**
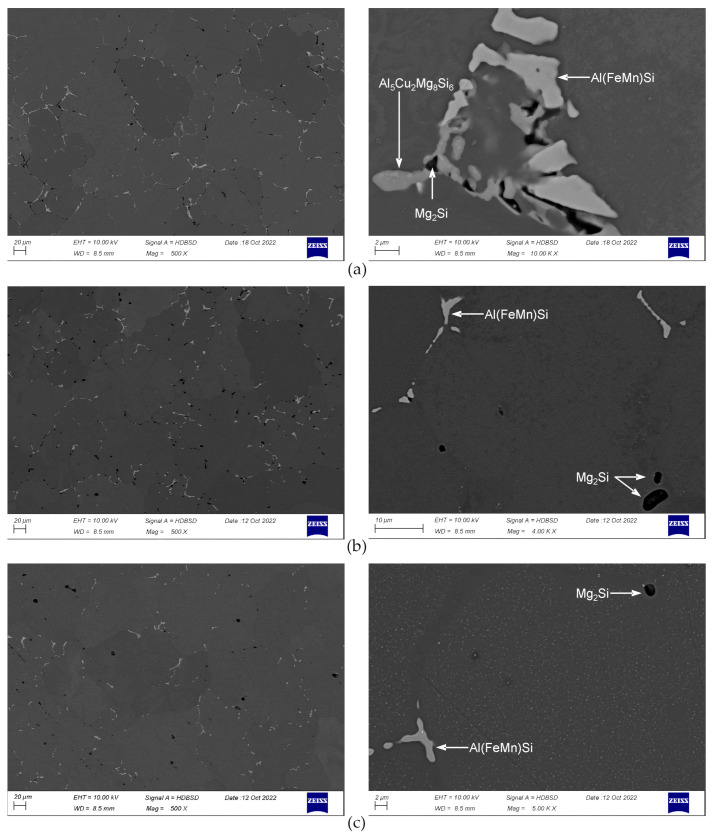
Microstructure of the alloy after heating to 475 °C for 10 h and soaking: (**a**) at 475 °C for 2 h; (**b**) at 475 °C for 2 h + 530 °C for 2 h; and (**c**) at 475 °C for 2 h + 530 °C for 2 h + 545 °C for 8 h. After complete soaking, samples were water quenched.

**Figure 5 materials-16-02091-f005:**
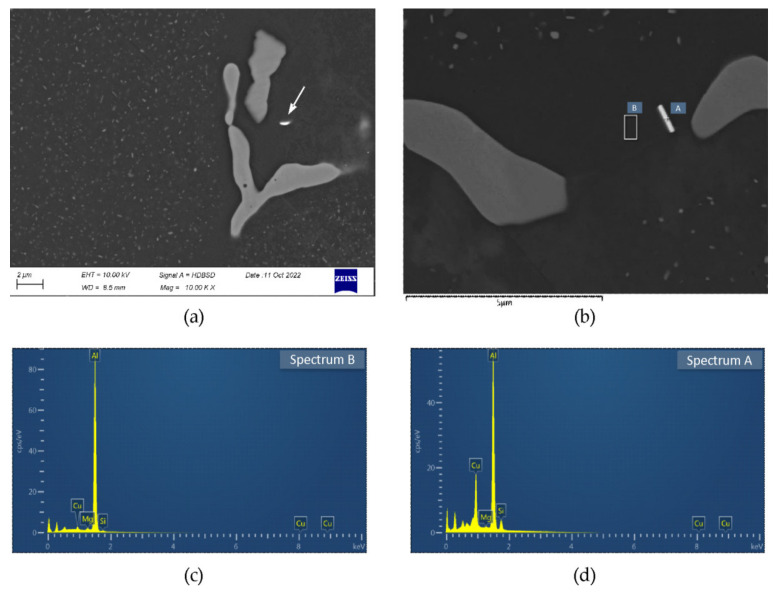
Alloy after homogenization with soaking at 475 °C for 2 h + 530 °C for 2 h + 545 °C for 8 h and water quenching: (**a**,**b**) examples of small particles rich in Cu and Si; (**c**,**d**) spectra from panel b.

**Figure 6 materials-16-02091-f006:**
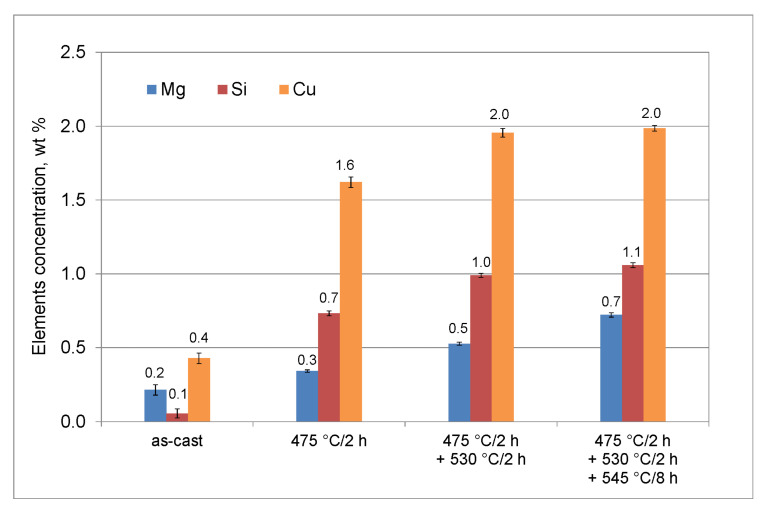
Concentration of Mg, Si, and Cu in dendrites/grains interiors in as-cast state and after each soaking stage (followed by water quenching). Error bars indicate standard deviation of the mean.

**Figure 7 materials-16-02091-f007:**
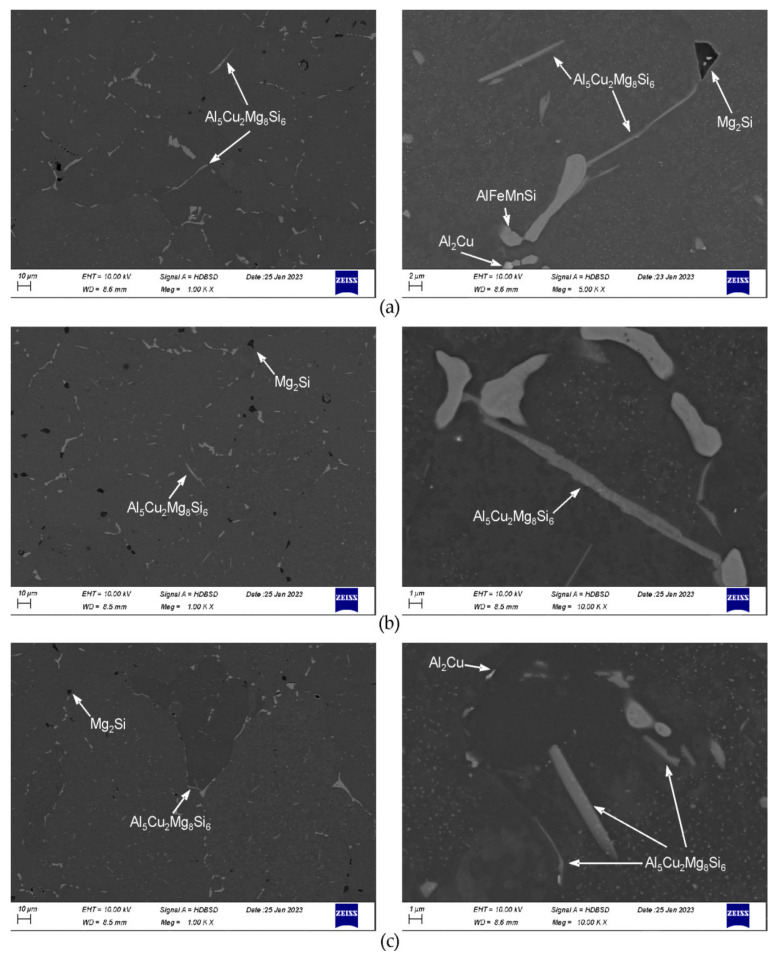
Microstructure after final soaking at 545 °C for 8 h and cooling at: (**a**) 50 °C/h; (**b**) 300 °C/h, and (**c**) 500 °C/h.

**Figure 8 materials-16-02091-f008:**
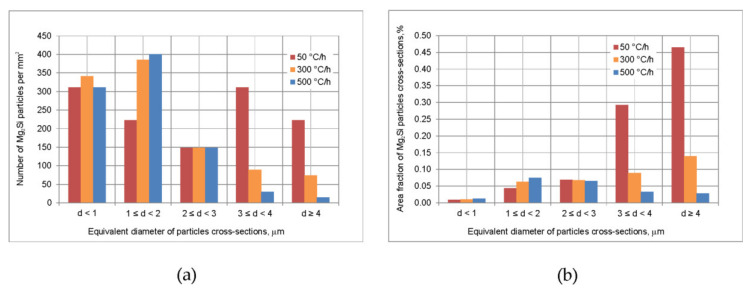
Effect of cooling rate from homogenization temperature on: (**a**) number of β-Mg_2_Si particles per mm^2^ and (**b**) area fraction of β-Mg_2_Si particles cross-sections.

**Figure 9 materials-16-02091-f009:**
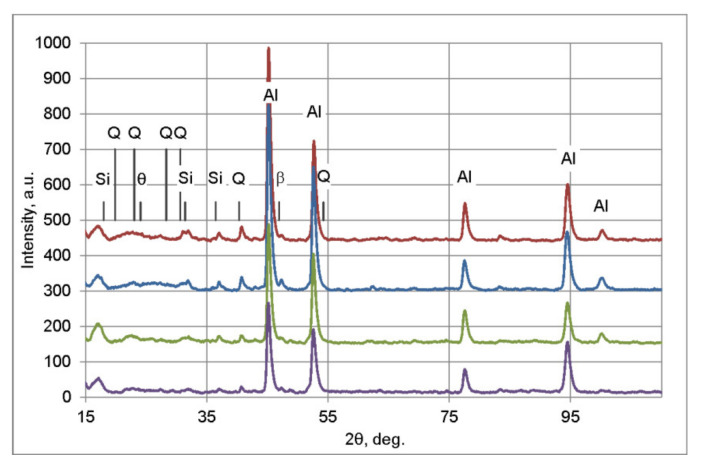
Results of X-ray diffraction phase analysis in as-cast state and after each soaking stage (followed by water quenching). The following JCPDS card was used for peaks identification: 00-004-0787, 00-034-0458, 01-075-0841, 00-041-1068, 03-065-26.95.

**Table 1 materials-16-02091-t001:** The chemical composition of investigated alloy, mass percentage.

Si	Fe	Cu	Mn	Mg	Cr	Ti	Zr
1.22	0.05	1.41	0.62	0.80	0.38	0.02	0.15

**Table 2 materials-16-02091-t002:** DSC test results.

State	Peak A		Peak B		Peak C		Bulk Melting
	Onset, °C	Heat, J/g	Onset, °C	Heat, J/g	Onset, °C	Heat, J/g	Onset, °C
As-cast	513.4	0.47	539.7	1.48	561.3	2.12	
475 °C/2 h–wq			544.8	0.90	560.4	3.77	
475 °C/2 h + 530 °C/2 h–wq					561.3	2.36	
475 °C/2 h + 530 °C/2 h + 545 °C/2 h–wq					562.2	0.79	
475 °C/2 h + 530 °C/2 h + 545 °C/4 h–wq					561.6	^1^	
475 °C/2 h + 530 °C/2 h + 545 °C/8 h–wq							573.4
475 °C/2 h + 530 °C/2 h + 545 °C/8 h–cooling 50 °C/h			542.7	0.02	560.3	5.95	
475 °C/2 h + 530 °C/2 h + 545 °C/8 h–cooling 300 °C/h					558.9	0.78	
475 °C/2 h + 530 °C/2 h + 545 °C/8 h–cooling 500 °C/h			546.7	0.10	562.9	0.14	

^1^ Peak C is merged with bulk melting peak and its heat was not determined.

## Data Availability

Not applicable.

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
