# Peer review of "Homogenization of Extrusion Billets of a Novel Al-Mg-Si-Cu Alloy with Increased Copper Content"

_materials, 2023, doi:10.3390/ma16052091_

Round 1
Reviewer 1 Report
The paper is well written, the text is clear and easy to read. The conclusions are in line with the evidence and arguments presented. . Adequate methods were used, the reached results are adequate presented, only in the fig. 1 magnification is missing (in the top right)- has not good quality.
The technological conditions for the preparation of the semi-finished product are not described.
The method of determining the temperature regimes of heat treatment is not described.
What is the accuracy of determining the content of the elements (the differences are relatively small)?
I believe that the diameters of the particles were not measured, but only the diameters of their sections (on the metallographic cut).
Reviewer 2 Report
Woźnicki et al. studied homogenization of extrusion billets of a novel Al-Mg-Si-Cu alloy with increased copper content. The work is interesting and can be considered for acceptance after proper revisions.
1. "The aim of the work was analysis of billets homogenization conditions enabling maximum dissolution of soluble phases during heating and soaking as well as their re-precipitation during cooling in form of particles capable for rapid dissolution during subsequent processes." The authors should describe exactly what characteristics the sample should have when optimal conditions or effects are achieved.
2. It is suggested that the author use the professional software Origin to draw all the graphs in the article.
3. In Figures 3 and 5, the subgraph number should be added.
4. The author's calibration towards the precipitated phase of existing products is basically based on the previous experience and EDS point analysis results. This is not only lack of diversity of detection methods, accuracy, and lack of scientificity. It is suggested that the authors provide more evidence such as TEM, SAED and Mapping, to make the conclusion more solid.
5. How to show the efficacy of achieving homogenization of extrusion billets? A series of subsequent work should be discussed and introduced by the author.
6. In Figure 9, JCPDS card numbers for XRD should be provided.
7. It is suggested to cite more relevant papers published in the last two years.
[1] Materials 2022, 15, 1206. https://doi.org/10.3390/ma15031206
[2] Journal of Alloys and Compounds 2021, 889, 161677. https://doi.org/10.1016/j.jallcom.2021.161677
[3] Rare Metals 2022, 41, 3546–3551. http://doi.org/10.1007/s12598-016-0706-7.
Reviewer 3 Report
The work focuses on analysis of billets homogenization environments enabling maximum dissolution of soluble phases during heating and soaking and on their re-precipitation during cooling in form of particles capable for rapid dissolution during billets preheating and extrusion. The article is very interesting. However, the following points needs to be answered in the revised manuscript.
1. Authors mentioned that The material was homogenized in laboratory conditions. What are the laboratory conditions?
2. Table 1. The chemical composition of investigated alloy, mass percentage. What is here investigated alloy? provide reference.
3. In section 3.1 provide full form of DSC?
4. Add few more points on Figure 6.
5. Add few more literatures from the year 2020, 2021, and 2022
6. Write key literature summary before the objective of the research.
Round 2
Reviewer 1 Report
Fig. 3c overlaps the calibration line in Fig. 3b!
On the metallographic cut you can not see diameters of the particles , but only the diameters of their sections.
Other comments were sufficiently accepted.
Reviewer 2 Report
The authors modified the article according to the reviewer's suggestion. However, the existing characterization of the paper does not match the level of the journal. More evidence such as TEM, SAED and Mapping should be provided to make the conclusion more solid.
Reviewer 3 Report
The paper now accepted and recommended for publications.
Author Response
Thank you once again for the review, valuable comments and suggestions and acceptanceof the revised paper.